# A Joint Training Framework for Open-World Knowledge Graph Embeddings

**Karthik Venkat Ramanan**                    VKARTHIK2019@GMAIL.COM
**Beethika Tripathi**                         BEETHIKA.TRIPATHI@GMAIL.COM
**Mitesh M. Khapra**                          MITESHK@CSE.IITM.AC.IN
**Balaraman Ravindran**                       RAVI@CSE.IITM.AC.IN
*Robert Bosch Centre for Data Science and AI, Indian Institute of Technology Madras,*
*Chennai, TN 60036, India*

## Abstract

Knowledge Graphs(KGs) represent factual information as graphs of entities connected by relations. Knowledge graph embeddings have emerged as a popular approach to encode this information for various downstream tasks like natural language inference, question answering and dialogue generation. As knowledge bases expand, we are presented with newer (open-world) entities, often with textual descriptions. We require techniques to embed new entities as they arrive using the textual information at hand. This task of open-world KG completion has received some attention in recent years. However, we find that existing approaches suffer from one or more of four drawbacks – 1) They are not modular with respect to the choice of the KG embedding model 2) They ignore best practices for aligning two embedding spaces 3) They do not account for differences in training strategy needed when presented with datasets with different description sizes and 4) They do not produce entity embeddings for use by downstream tasks. To address these problems, we propose FO1K (**F**ramework for **O**pen-Wor**l**d **K**G embeddings) - a technique that jointly learns embeddings for KG entities from descriptions and KG structure for open-world knowledge graph completion. Additionally, we modify existing data sources and make available YAGO3-10-Open and WN18RR-Open two datasets that are well suited for demonstrating the efficacy of open-world KG completion approaches. Finally, we empirically demonstrate the effectiveness of our model in improving upon state-of-the-art baselines on several tasks resulting in performance increases of up to 72% on mean reciprocal rank.

## 1. Introduction

Knowledge graphs (KG) are facts structured in the form of a graph, where the nodes are the entities, and the edges are the relationships between those entities. For example, consider the fact "Narendra Modi is the Prime Minister of India and lives in India.". In this example, `Narendra Modi` and `India` are two entities connected by relations *isPrimeMinister* and *livesIn*. KGs are often represented in the form of triples, example, (`Narendra Modi`, *isPrimeMinister*, `India`). KG embeddings are continuous multi-dimensional vectors of entities and relations that, through a predefined scoring function, rank real facts in a knowledge base over spurious ones. KG embeddings have a number of applications ranging from KG completion [Bordes et al., 2013] to natural language inference [Peters et al., 2019], knowledge-aware conversation generation [Zhou et al., 2018], question-answering [Saxena et al., 2020] and recommendation [Chen et al., 2019].

Large knowledge bases like DBpedia [Auer et al., 2007] rely on unstructured information sources like Wikipedia, which is growing at the rate of 602 new articles per day[1]. Applications or tasks that rely on knowledge graph embeddings require approaches to embed new entities introduced due to evolving knowledge bases. Traditional (or closed-world) KG embedding models, are limited to the transductive settings, where all the entities and the relations are known during learning. Thus, these approaches are not suitable for KGs where the entities and the relations are augmented over time. An ideal KG embedding model should be able to generalise to newer (or open-world) entities and relations in the KGs without re-learning on the updated KGs. We require a method to embed any new (or open-world) entities based on the rich semantic information that exist about them in the form of new articles, and also based on the existing connections in the KG. This brings up two tasks that make use of an entity's description- (1) *Open-World KG Completion*- which is the task of relating the new entity to the rest of the KG, and (2) *Open-World Entity Embedding*- which is the task of producing an embedding for a new entity.

Few approaches have been introduced in recent years for these open-world KG related tasks. Studying the literature, we identify five critical properties that make an open-world KG embedding approach efficient, robust and useful for a plethora of applications.

1. **Open-world Embedding Generation:** KG applications like question answering or recommendation require entity and relation embeddings. Techniques like ConMask [Shi and Weninger, 2018] which are only optimised for KG Completion and do not produce entity/relation embeddings cannot be used for such downstream applications. Hence, it is important that the approaches must also embed open-world KG entities.

2. **Joint Training:** To embed open-world entities using their descriptions, we need to align two spaces – description space and KG structure space. Techniques that learn this alignment while keeping one or both of these spaces fixed are said to be *offline*. Techniques that learn the alignment operation and individual embeddings at the same time are called *joint*. Studies [Ormazabal et al., 2019] have shown that embeddings produced by joint training of the two spaces have significantly improved geometric properties as compared to offline training. These geometric properties are important to retain the KG structure on the introduction of new entities, thus, resulting in better performance in downstream applications.

3. **Efficient Ranking:** Connecting a new entity to the rest of the KG requires the ranking of potential connections. The score is generated for all possible triples which are ranked to determine the top connections [Bordes et al., 2013, Yang et al., 2015, Trouillon et al., 2016]. The scoring function should preferably have time complexity of the order of the embedding dimensions. For real-time applications, the time and memory requirements for inference must be minimal and thus large neural networks models are not efficient as a scoring function to score all possible triples.

4. **Modular:** As observed by [Chang et al., 2020], different KG embedding approaches are suitable for different applications. For instance, translational methods [Bordes et al., 2013] are better at clustering and entity classification applications, whereas multiplicative methods [Trouillon et al., 2016, Yang et al., 2015] are better at KG completion. For this reason, our open-world framework must be modular with respect to the choice of the structural

---

1. https://en.wikipedia.org/wiki/Wikipedia:Statistics

embeddings to accommodate a variety of applications.

5. **Sequence-Size Aware:** Any proposed approach must be able to leverage long descriptions using state-of-the-art language models. At the same time, large, popular KGs like Wikidata [Vrandečić and Krötzsch, 2014] have concise sentence-long descriptions that do not benefit from large language models. Any proposed approach must be efficient and perform competitively when run on datasets with short descriptions.

Our contributions are summarised below. Our code and datasets are available at https://github.com/RBC-DSAI-IITM/KnowledgGraphZeroShotLearning.

1. We propose FO1K - the first-ever **joint**, **modular**, **efficient** and **sequence size aware** framework to produce open-world KG entity **embeddings**.
2. We empirically evaluate performance of our model on open-world KG completion, closed-world KG completion and entity classification. We demonstrate significant improvement of upto 72% on mean reciprocal rank over the state-of-the-art.

## 2. Problem Definition

Before discussing existing approaches and our solution, we give a brief introduction to the problem of learning embeddings for open-world KG entities. Knowledge graphs primarily consist of a set of entities $\mathcal{E} = \{e_1, e_2, ...e_{|\mathcal{E}|}\}$, a set of relations $\mathcal{R} = \{r_1, r_2, ...r_{|\mathcal{R}|}\}$ and a set of (head,*relation*,tail) triples $\mathcal{T} \subseteq \{\tau_i : (h_i, r_i, t_i)|h_i, t_i \in \mathcal{E}, r_i \in \mathcal{R}\}$. We refer to the set of entities $\mathcal{E}$ with relationship information in the existing KG as the *closed* or *structural* entity set. In many real world scenarios we often have the descriptions of each entity in our knowledge graph. We denote the description set as $\mathcal{D} = \{d_1, d_2, ...d_{|\mathcal{E}|}\}$. $\boldsymbol{e_i}$ and $\boldsymbol{r_i}$ are real (or complex valued) vectors or embeddings. For model like TransE [Bordes et al., 2013] or DistMult [Yang et al., 2015], $\boldsymbol{e_i}, \boldsymbol{r_i} \in \mathbb{R}^\delta$ (where $\delta$ denotes the embedding dimension). $\boldsymbol{e_i}, \boldsymbol{r_i} \in \mathbb{C}^\delta$ in the case of models like ComplEx [Trouillon et al., 2016]. Throughout we use boldface to indicate the vector embedding corresponding to an entity or relation. We use $\boldsymbol{h}$, $\boldsymbol{r}$ and $\boldsymbol{t}$ to denote head, relation and tail embeddings respectively, in a triple. Entities in a knowledge graph may appear as heads, tails or both. As knowledge bases expand we are presented with a new set of entities $\mathcal{E}' = \{e_1', e_2', ...e_{|\mathcal{E}'|}'\}$ and their corresponding descriptions $\mathcal{D}' = \{d_1', d_2', ...d_{|\mathcal{E}'|}'\}$. Our task is to embed the open entity set $\mathcal{E}'$ using $\mathcal{D}'$.

## 3. Related Work

In this section, we briefly summarise state of the art closed-world or structural KG embedding approaches, approaches with open-world entities and their limitations.

**Structural KG Embeddings:** These models use only the triple or structural information of a KG. Most models are learnt by minimising an energy function on $(\boldsymbol{h}, \boldsymbol{r}, \boldsymbol{t})$ triples through negative sampling. TransE [Bordes et al., 2013] uses a translational energy function whereas DistMult [Yang et al., 2015] and ComplEx [Trouillon et al., 2016] use multiplicative energy functions. The energy and loss functions of all three methods are detailed in Table 8.

**Approaches with open-world entities:** These models use additional textual descriptions of the entities along with structural information available in KG.(1) **DKRL** [Xie et al., 2016], constructs two separate spaces - the structural embedding space and the description embedding space and aligns the two using a version of the cross-completion loss we discuss

in Section 6.3. A key limitation of this approach is its inability to generate embeddings for new open-world entities as a consequence of training the two embeddings in separate spaces. (2) **JointE** [Zhao et al., 2017], uses tf-idf weighted bag-of-words features to encode a description and proposes a custom multiplicative energy function to score a triple. The approach is not modular hence limits the usability of approach for specific applications. Similarly, (3) **ConMask**'s [Shi and Weninger, 2018] model also uses a custom energy function to score a triple. A triple is scored using a combination of multiple features generated using masked fusion of the entity description with entity and relation names. The usage of a large neural network model to score a triple limits the scalability of the model to score all possible triples, thus filtering methods are used to filter out certain triples resulting in information loss. Further details to be discussed in Sections 6.2 and 6.3. (4) **OWE**, uses offline training to learn a projection matrix to align descriptions to KG structure. The KG embeddings are pre-trained and not modified at alignment time. We discuss the limitations of such an approach in Section 6.3. We have compared the properties of various approaches and our framework in Table 9.

## 4. Proposed `FO1K` Model

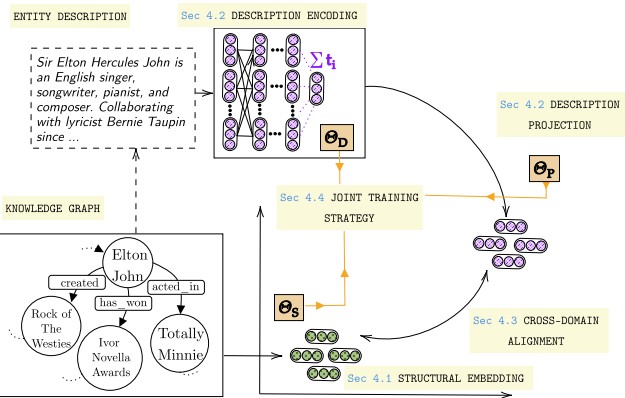

Figure 1: An illustration of `FO1K`'s architecture

Overall architecture of our framework, `FO1K`, is presented in Figure 1. It has three components- (1) Structural Embedding module to learn the embeddings of the entities and relations already existing in a KG (2) Description Embedding module which is an encoder to represent the textual information of open-world as well as closed-world entities. (3) Description Projection module to project the description to a space where the structure and description embeddings are aligned, this represents the KG embedding space used for any of the downstream tasks like KG completion or entity classification. When presented with a new entity, we induce the KG embedding by encoding it's description and projecting it to KG embedding space. The following sections will discuss these modules in detail along with various training strategies that we have used.

### 4.1 Structural Embeddings

Our framework is agnostic to the choice of the structural KG embedding model. Unless otherwise specified, we use ComplEx as our default method as it has out-performed other structural embedding methods as reported in a recent survey [Dai et al., 2020].

We briefly explain a few notations first. Let $\mathcal{E}^{test}$ and $\mathcal{E}^{valid}$ be the open-world test and validation sets of entities that are not available in the training set. We denote the train, validation and test splits of the triple set $\mathcal{T}$ by $\mathcal{T}_{train}$, $\mathcal{T}_{valid}$ and $\mathcal{T}_{test}$. $\mathcal{T}_{test}^{open}$ and $\mathcal{T}_{valid}^{open}$ denote the test and validation sets of triples where either the head or tail of each triple occurs in $\mathcal{E}^{test}$ or $\mathcal{E}^{valid}$ respectively. We augment the train split $\mathcal{T}_{train}$ with corrupted triplets, i.e, $\mathcal{T}_{corr} \subseteq \{\tau_{corr.i} : (h_i, r_i, t_i)|h_i, t_i \in \mathcal{E}, r_i \in \mathcal{R}, (h_i, r_i, t_i) \notin \mathcal{T}_{train}\}$ to obtain $\mathcal{T}^{\circ} = \mathcal{T}_{train} \cup \mathcal{T}_{corr}$. We use the indicator variable $I_{h,r,t} \in \{1, -1\}$ to return 1 (or -1) corresponding to whether $(h, r, t) \in \mathcal{T}_{train}$ (or not). We denote the parameters of this module (the real or complex embeddings) by $\mathbf{\Theta_S}$. The loss function used by ComplEx is represented in equation 1, where $Re(x)$ is real part of complex value x and $\bar{t}$ is complex conjugate. The loss functions of TransE and DistMult are shown in Table 8. We also include a regularization loss term $\mathscr{L}_{reg}$ which is the average squared $\ell_2$-norm of all the structural embeddings with the regularization parameter $\lambda$.

$$\mathscr{L} = \sum_{(h,r,t)\in\mathcal{T}^{\circ}} log(1 + exp(-I_{h,r,t}Re(\langle\mathbf{h}, \mathbf{r}, \bar{\mathbf{t}}\rangle))) \tag{1}$$

### 4.2 Induced Embeddings

To induce new embeddings for open-world entities, we learn to encode and project entity descriptions to KG entity space at train time.

**Description Encoding:** To make our framework sequence-size-aware, we use different text embedding approaches based on the description size to encode entity descriptions. For short entity descriptions, ten or fewer words, we use the Continuous Bag-of-words (CBOW) encoder which assumes that similar entities have related keywords. The representation of the entity, $\mathbf{\Theta_{D_s}}$ (s for *short* description), is determined by the summation of all words' pre-trained word2vec [Mikolov et al., 2013] embeddings. For long entity descriptions we need to capture more complex interactions. Hence we use the transformer-based model RoBERTa [Devlin et al., 2018] which uses attention or differential weighing of the different parts of the description to capture the most significant information. To the best of our knowledge, FO1K is the first framework to use a transformer-based encoder to embed open-world entities. RoBERTa tokenizes each input sentence. The output at the final layer is a 768-dimensional vector for each token. The entity description $\mathbf{\Theta_{D_l}}$ (l for *long* description) is generated by taking the average of all the tokens' embeddings.

**Description Projection:** We use an affine transformation function to project the entity's description embedding $\phi_D(e)$ to the KG embedding space, represented in equation 2. The projection parameters, $\mathbf{\Theta_P}$, are trained jointly, in tandem with the rest of the model. $M$ and $b$ project the description embedding to to the real space and $M_i$ and $b_i$ project to the imaginary space. Unlike ComplEx, if the embedding space only has a real component as in TransE or DistMult, then we use just the real transformation matrix and bias.

$$\phi_P(e) = M\phi_D(e) + b + i(M_i\phi_D(e) + b_i) \tag{2}$$

### 4.3 Cross-Domain Alignment

For cross-domain alignment, we minimise the squared $\ell_2$ distance between projected description embeddings $\boldsymbol{\phi_P}(\boldsymbol{\phi_D}(\boldsymbol{e}))$ and structural embeddings $\mathbf{e}$ as in equation 3. We borrow best practices for aligning the two spaces from Bilingual Lexicon Induction (BLI) literature, where this is a widely explored problem. It aims to learn an alignment between the word vector spaces of two languages. A common practice in BLI is to use a combination of signals of varying scales for alignment at the level of parallel words, sentences and documents. [Chandar et al., 2014] and [Gouws et al., 2015] demonstrate that sentence-level alignment is sufficient and does not require supervision at paired word level for competitive BLI performance. [Klementiev et al., 2012] train bilingual embeddings by jointly using sentences to learn a language model on each corpus and enforcing word alignments using a parallel dictionary. Most relevant to our work is BiSkip, [Luong et al., 2015] which uses a skipgram objective where the probability of a target language word given the source language context is maximised and vice versa. We find that including such cross-modality objectives in addition to vanilla projection loss consistently improves performance. BiSkip uses parallel sentences to apply a bilingual skipgram objective to parallel words. We can treat an entity's immediate graph neighbourhood as its context and perform a similar operation. The loss term $\mathscr{L}_X$ (X for cross) is represented in equation 4. We call this the cross-domain KG completion term. The overall loss function is represented in Equation 5, where $\alpha$ and $\beta$ are hyperparameters. We demonstrate the effectiveness of the cross-domain terms empirically in Table 6 and Section 6.3.

$$\mathscr{L}_{proj} = \sum_{e \in \mathcal{E}} ||\boldsymbol{\phi_P}(\boldsymbol{\phi_D}(\boldsymbol{e})) - \boldsymbol{e}||_2 \tag{3}$$

$$\mathscr{L}_X(h, r, t) = \mathscr{L}(\boldsymbol{h}, \boldsymbol{r}, \boldsymbol{\phi_P}(\boldsymbol{\phi_D}(\boldsymbol{t}))) + \mathscr{L}(\boldsymbol{\phi_P}(\boldsymbol{\phi_D}(\boldsymbol{h})), \boldsymbol{r}, \boldsymbol{t}) \tag{4}$$

$$\mathscr{L}_{\texttt{FOlK}} = \overbrace{\mathscr{L} + \lambda\mathscr{L}_{reg}}^{\text{closed-world completion}} + \overbrace{\alpha\mathscr{L}_{proj} + \beta\mathscr{L}_X}^{\text{open-world alignment}} \tag{5}$$

### 4.4 Training Strategy

Utilising a large contextual model like RoBERTa necessitates an appropriate training strategy. This is because RoBERTa takes a few hours per epoch to fine-tune and converges in around 2-3 epochs for long descriptions, while structural KG embedding methods takes a few minutes per epoch and requires anywhere between 100-400 epochs to converge. So we adopt a novel *phase-wise training approach* to train all three modules. We optimise the same loss term $\mathscr{L}_{\texttt{FOlK}}$ but separate the training of the closed-world embedding module ($\boldsymbol{\Theta_S}$) and the description encoding and projection modules ($\boldsymbol{\Theta_D}$ and $\boldsymbol{\Theta_P}$) into separate phases, each trained to convergence. Algorithm 1 details this procedure. We denote this model FOlK(l), the l standing for long descriptions.

We briefly describe a simpler modification of FOlK(l). When entities only have short descriptions, there is limited contextual information to be gleaned. In such cases, averaging the Word2Vec word embeddings[2] in a description is an effective way to encode semantic information and using larger architectures does not impart a performance benefit (Table 2).

---

2. https://wikipedia2vec.github.io/wikipedia2vec/pretrained/

In this case, we minimise the objective $\mathscr{L}_{\texttt{FO1K}}$ (Eqn. 5) using batch-wise stochastic gradient descent, with $\boldsymbol{\Theta_S}$ and $\boldsymbol{\Theta_P}$ initialised at random. We minimise $\mathscr{L}_{\texttt{FO1K}}$ training all 3 sets of parameters - $\boldsymbol{\Theta_S}$, $\boldsymbol{\Theta_{D_s}}$ and $\boldsymbol{\Theta_P}$.

$$\underset{\Theta_S \Theta_{D_s} \Theta_P}{\operatorname{argmin}} \mathscr{L}_{\texttt{FO1Kx}} \tag{6}$$

---

**Algorithm 1:** Algorithm for $\texttt{FO1K}(l)$

---

**Input:** Triplets: $\mathcal{T}_{train}$, $\mathcal{T}_{valid}$, $\mathcal{T}_{valid}^{open}$
**Output:** $\boldsymbol{\Theta_S}, \boldsymbol{\Theta_{D_l}} \boldsymbol{\Theta_P}$
Initialise $\boldsymbol{\Theta_{D_l}}$ on RoBERTa's pre-training tasks
Initialise $\boldsymbol{\Theta_S}$ and $\boldsymbol{\Theta_P}$
**while** *MRR no longer improves on* $\mathcal{T}_{valid}$ **do**
   | Train $\boldsymbol{\Theta_S}$ by optimising $\mathscr{L} + \lambda\mathscr{L}_{reg}$;
**end**
$i \leftarrow 0$
**while** *MRR no longer improves on* $\mathcal{T}_{valid}^{open}$ **do**
   Phase 1: Freeze $\boldsymbol{\Theta_{D_l}}$ and $\boldsymbol{\Theta_P}$
   **if** $i \neq 0$ **then**
      | Train $\boldsymbol{\Theta_S}$ by optimising $\mathscr{L} + \lambda\mathscr{L}_{reg} + \alpha\mathscr{L}_{proj} + \beta\mathscr{L}_X$; // Until MRR no
            longer improves on $\mathcal{T}_{valid}$
   **end**
   Phase 2: Freeze $\Theta_S$
   Train $\boldsymbol{\Theta_{D_l}}$ and $\boldsymbol{\Theta_P}$ by optimising $\alpha\mathscr{L}_{proj} + \beta\mathscr{L}_X$; // //Until MRR no longer
       improves on $\mathcal{T}_{valid}^{open}$
   $i \leftarrow i + 1$;
**end**

---

## 5. Datasets

We have evaluated our model on following datasets: **FB15k-237-OWE** [Shah et al., 2019] & **FB15k-237-OWE(L):** FB15k-237-OWE paired with descriptions from Freebase which are of longer lengths, is called the FB15k-237-OWE(L) dataset. **YAGO3-10-Open:** YAGO3-10 [Dettmers et al., 2018] is a subset of the YAGO[3] knowledge base. We use DBpedia to extract long descriptions of the entities. **WN18RR-Open** WN18RR [Dettmers et al., 2018] is derived from WordNet. We use the synset definitions from WordNet as descriptions. For the statistics of our datasets please refer to Table 1.

**DSA Score**: We report a simple heuristic to measure the relative difficulties of aligning the description and structure embedding spaces across datasets. The Description-Structure Alignment (DSA) score is a measure of the pre-existing alignment between the descriptions of the entities and the KG structure embeddings. For every entity in $\mathcal{E}$, we train a set of structural KG embeddings independently, without any reference to the descriptions of the entities. From these structural embeddings we construct a nearest-neighbour graph for the entities in $\mathcal{E}$ using $\ell_2$ distance, where each node has an out-degree of 100. We denote the edge set of this graph as $ES_1$. For every entity in $\mathcal{E}$, we also average the word2vec embeddings of

---

3. https://yago-knowledge.org/

Table 1: Dataset Statistics. $\overline{|d|}$ is the average entity description length. $|c.c|$ is the number of connected components in the train graph. DSA is the Description-Structure Alignment score. Please refer to Section 5 for more details.

| | Entities Statistics | | | | | | Relationships Statistics | | | | |
|---|---|---|---|---|---|---|---|---|---|---|---|
| Dataset | $|\mathcal{E}|$ | $|\mathcal{E}'|$ | $|\mathcal{R}|$ | $\overline{|d|}$ | DSA | $|c.c|$ | $|\mathcal{T}_{train}|$ | $|\mathcal{T}_{test}^{open}|$ | $|\mathcal{T}_{valid}^{open}|$ | $|\mathcal{T}_{test}|$ | $|\mathcal{T}_{valid}|$ |
| FB15k-237-OWE | 12324 | 2081 | 235 | 4.9 | 12.4 | 2 | 242489 | 22393 | 9424 | 7806 | 5000 |
| FB15k-237-OWE(L) | | | | 138.7 | 12.6 | | | | | | |
| YAGO3-10-Open | 107327 | 11443 | 32 | 94.0 | 4.4 | 1 | 769694 | 56334 | 37557 | 39794 | 26533 |
| WN18RR-Open | 36119 | 4824 | 11 | 12.9 | 4.9 | 1 | 69122 | 4962 | 3308 | 3133 | 2086 |

the words in the entity's description. As in the case of the strucutral embeddings, we then construct a nearest neighbour graph for the entities from this set of description embeddings and denote it's edge set by $ES_2$. The DSA score is the edge-overlap between the two graphs, divided by the number of nodes: $|ES_1 \cap ES_2|/|\mathcal{E}|$. A higher value indicates better initial alignment, and therefore an easier alignment task.

## 6. Experiments

Through our experiments, we address the following research questions:

- **RQ1:** What is the quality of the open-world embeddings generated by FO1K?
- **RQ2:** How well does FO1K satisfy the properties outlined in Section 1?
- **RQ3:** What are the contributions of each of our model components?

### 6.1 Training

We train both FO1K(s) and FO1K(l) through batch-wise stochastic gradient descent (SGD). For FO1K(s), the learning rate and negative sample rate are the same as the best parameters for the structural model. For information on the hyperparameters of the structural model, please refer to the additional material. Training the structural model by minimising $\mathbf{\Theta_S}$ on $\mathscr{L} + \lambda\mathscr{L}_{reg}$ is an initialisation step for FO1K(l). In phase 2 of FO1K(l), we train $\mathbf{\Theta_{D_l}}$ and $\mathbf{\Theta_P}$ using Adam with a learning rate of 0.00005. In phase 1, $\mathbf{\Theta_S}$ is trained using the same hyperparameters used by the structural model. $\beta$ is always set to 1. For both FO1K(s) and phase 1 of FO1K(l), we find that an optimal value of $\alpha$, 0.001 works well across all datasets. It was identified through grid search in the space of [1, 0.1, 0.01, 0.001]. During phase 2 of FO1K(l), $\alpha$ is always set to 1. Wherever we report results of FO1K(l), we also indicate the performance improvement across iterations.

### 6.2 Experimental Setup

We study our research questions using three tasks – (1) Open-World KG Completion (2) Closed world KG Completion and (3) Entity Classification The details are discussed here. **Open-World KG Completion and Closed-World KG Completion:** For the KG completion tasks (Tables 2, 3, 4, 12), we follow the filtered ranking procedure introduced by [Bordes et al., 2011]. In *filtered* ranking, we remove the correct answers that occur in the training set from $\mathcal{E}$ before performing the ranking operation. That is, when ranking tails for $(h_i, r_i)$ we remove all $\{t|(h_i, r_i, t) \in \mathcal{T}_{train}\}$ from $\mathcal{E}$.

Table 2: Comparing open-world KG completion results on FB15k-237-OWE, a dataset with short description.[4]

| | FB15k-237-OWE | | | |
|---|---|---|---|---|
| Model | MRR | H@1 | H@3 | H@10 |
| JointE | 6.7 | 2.5 | 7.0 | 14.2 |
| DKRL-CNN | 19.0 | 13.0 | 21.2 | 31.0 |
| DKRL-CBOW | 19.3 | 13.1 | 21.5 | 31.9 |
| ConMask | 9.1 | 3.7 | 9.5 | 20.5 |
| OWE | 35.2 | 27.8 | 38.6 | 49.1 |
| FO1K(l) iter. 1 | 38.8 | 29.9 | 42.6 | 54.5 |
| FO1K(l) iter. 2 | **39.1** | **32.1** | 42.5 | 52.1 |
| FO1K(s) | **39.1** | 30.3 | **43.0** | **56.1** |

Table 3: Performance comparison on closed-world KG completion on FB15k-237-OWE.

| | FB15k-237-OWE | | | |
|---|---|---|---|---|
| Model | MRR | H@1 | H@3 | H@10 |
| ComplEx | 35.6 | 24.3 | 41.2 | 57.3 |
| FO1K(s) | **38.1** | **26.5** | **44.3** | **61.6** |

Table 4: Performance comparison on open-world KG completion across various datasets with long descriptions. The results reported are for filtered evaluation (not target-filtered)[4].

| | YAGO3-10-Open | | | | WN18RR-Open | | | | FB15k-237-OWE(L) | | | |
|---|---|---|---|---|---|---|---|---|---|---|---|---|
| Model | MRR | H@1 | H@3 | H@10 | MRR | H@1 | H@3 | H@10 | MRR | H@1 | H@3 | H@10 |
| JointE | 5.1 | 1.8 | 4.5 | 11.0 | 8.2 | 4.5 | 8.2 | 16.0 | 10.3 | 5.1 | 10.9 | 20.0 |
| DKRL-CNN | 2.6 | 1.5 | 2.2 | 4.1 | 2.5 | 1.1 | 2.4 | 5.1 | 19.9 | 13.9 | 21.7 | 32.1 |
| DKRL-CBOW | 2.7 | 1.6 | 2.4 | 4.2 | 2.4 | 1.0 | 2.3 | 4.9 | 20.7 | 14.5 | 22.6 | 33.4 |
| ConMask | 17.3 | 10.3 | 18.9 | 31.3 | 23.3 | 10.3 | 22.7 | 38.4 | 21.1 | 14.0 | 23.4 | 34.6 |
| OWE | 21.6 | 14.9 | 23.3 | 34.3 | 21.7 | 17.3 | 23.4 | 29.4 | 32.4 | 25.1 | 35.6 | 46.0 |
| FO1K(l). iter. 1 | 25.7 | 19.0 | 27.5 | 38.9 | 35.6 | 30.9 | 37.9 | 45.5 | 42.4 | 33.6 | 45.7 | 57.2 |
| FO1K(l). iter. 2 | **26.5** | **19.5** | **28.0** | **40.0** | **40.3** | **32.2** | **40.8** | **50.0** | **43.6** | **34.8** | **47.6** | **59.8** |

**Entity Classification:** We follow the same procedure as in DKRL[Xie et al., 2016] to identify entity type information from Freebase for classification. We set up a classification task on the embeddings by training one-vs-rest logistic classifiers on entities in $\mathcal{E}$. We then test the performance of our model on the open-world embeddings generated for entities in $\mathcal{E}'$. We report micro-f1 and macro-f1 scores in Table 5.

### 6.3 Analysis

**RQ1 - Open-World Embedding Quality:** The results of FO1K on open-world KG completion for datasets with short and long description are presented in Tables 2 and 4 respectively. FO1K significantly outperforms baselines with an average MRR improvement of 35% across all datasets and an improvement of 72% on WN18RR-Open. YAGO3-10-Open, FB15k-237-OWE and FB15k-237-OWE(L) (along with Dbpedia and FB20k used by other baselines) all have either entities or descriptions derived from Wikipedia. Hence, WN18RR-Open is an important dataset to compare with to evaluate the performance on KGs from other domains. RoBERTa's large-scale pre-training ensures that FO1K is fairly robust to such domain shifts. On the task of open-world entity classification also, we see dramatic improvements (Table 5), with a macro-f1 score improvement of 33%.

---

4. Please take note of the filtering used when comparing results across papers. In OWE, the results reported in Table 3 of the original paper are target-filtered. The results in Table 4 of the original paper are filtered. For reasons explained in Section 6.3 we only report filtered results in this work.

DKRL's poor performance (in Table 4) on YAGO3-10-Open and WN18RR-Open is due to (1) weak pre-existing alignment between the structure and description space, (2) challenges in scaling to large graphs. The alignment between structure and description space is represented as Description-Structure Alignment (DSA) scores in Table 1. WN18RR-Open and YAGO3-10-Open have DSA scores 60% and 65% lower than that of FB15k-237-OWE respectively, making them harder alignment problems. Also, the specifics of DKRL's training strategy make it unable to scale to large graphs.

The authors of ConMask have reported results using *target-filtered ranking*. Instead of ranking across all entities as tails, target-filtering reduces the search space by ranking only those entities as tails that have been associated with the test relation at train time, i.e. for a test $(h_i, r_i)$ we only rank $\{t|(h_j, r_i, t) \in \mathcal{T}_{train}, h_j \in \mathcal{E}\}$. This ranking approach not only limits the connections that can be discovered, but also obscures poor performance. We evaluated that in a target-filtered setting for ConMask we get similar results as those reported in OWE. But without target-filtering we see a large performance drop-off of 83% in Hits@1 (Table 2).

Table 5: Comparing open-world KG embedding techniques for the task of entity classification.

|  | FB15k-237-OWE(L) | |
|---|---|---|
| Model | macro-f1 | micro-f1 |
| OWE | 56.1 | 58.5 |
| FO1K(l). iter. 1 | 73.2 | 80.3 |
| FO1K(l). iter. 2 | **74.8** | **81.7** |

Table 6: Ablation Studies on FB15k-237-OWE(L)

| Model | MRR | H@1 | H@3 | H@10 |
|---|---|---|---|---|
| M1. FO1K(s) w/out $\beta\mathscr{L}_X$ | 33.6 | 25.6 | 36.9 | 49.0 |
| M2. FO1K(s) w/out $\alpha\mathscr{L}_{proj}$ | 34.5 | 26.2. | 37.1 | 50.9 |
| M3. FO1K(s) | 37.5 | 29.0 | 40.9 | 55.7 |
| M4. FO1K(l) w/out $\beta\mathscr{L}_X$ | 42.5 | 34.5 | 46.2 | 57.7 |
| M5. FO1K(l) w/out $\alpha\mathscr{L}_{proj}$ | 15.8 | 11.2 | 18.1 | 26.0 |
| M6. FO1K(l) | **43.6** | **34.8** | **47.6** | **59.8** |

Table 7: FO1K's performance on open-world KG completion, paired with different closed-world embedding approaches. OWE's results are taken directly from their paper. We use FO1K(s) as the dataset is FB15k-237-OWE.

|  | TransE | | | | DistMult | | | | ComplEx | | | |
|---|---|---|---|---|---|---|---|---|---|---|---|---|
|  | MRR | H@1 | H@3 | H@10 | MRR | H@1 | H@3 | H@10 | MRR | H@1 | H@3 | H@10 |
| OWE | 28.7 | 21.9 | 31.7 | 41.0 | 34.4 | 26.6 | 37.7 | 49.2 | 35.2 | 27.8 | 38.6 | 49.1 |
| FO1K(s) | **42.3** | **33.2** | **46.4** | **59.4** | **37.0** | **28.5** | **40.0** | **53.2** | **39.1** | **30.3** | **43.0** | **56.1** |

**RQ2 - Properties: (1)Ranking**: We computed the time taken to complete filtered ranking on $\mathcal{T}_{test}^{open}$ on an NVIDIA Tesla V100 with 32GB of VRAM. We observe, on average, a 22x speedup using simple embedding energy functions for scoring rather than large networks. ConMask's ranking is inefficient to enumerate every possible fact in a graph. For e.g., on YAGO3-10-Open ConMask takes 371 seconds, while FO1K takes 71 seconds. **(2) Joint Training:** Figure 2, demonstrates the visualisation of embeddings in 3D space using PCA. The embeddings induced using joint training (blue dots) and pre-existing closed-world embeddings (red dots) are homogeneous in embedding space, whereas those induced using OWE are clustered separately. This helps explain the significant improvement in open-world embedding performance using joint training. **(3) Embedding Generation:** The effect of generating embeddings for open-world entities is that we can perform downstream tasks on them without having to retrain our knowledge base. For instance, we can

not perform the open-world classification (Table 5) experiment on DKRL or ConMask as they do not generate embeddings for open-world entities. **(4) Modular:** From Table 7 we observe that FO1K can accomodate a variety of structural energy functions, obtaining an average MRR improvement of 23%. It is interesting to observe here that despite TransE having 8% improved open-world performance on MRR over ComplEx, ComplEx has superior closed-world performance (not shown in the table - MRR of 38.6 vs MRR of 26.5). This lends further support to allowing modular energy functions, based on the application. **(5) Sequence Size Aware:** The results presented in Tables 4 and 2 indicate that FO1K(l) performs competitively regardless of description length. Table 2 indicates that FO1K(s) has similar performance and is sufficient for SOTA results on datasets with short descriptions. We don't observe a performance benefit to using a CNN in DKRL (Tables 4, 2). This suggests that incorporating larger encoders is non-trivial and requires a more careful consideration of training strategy.

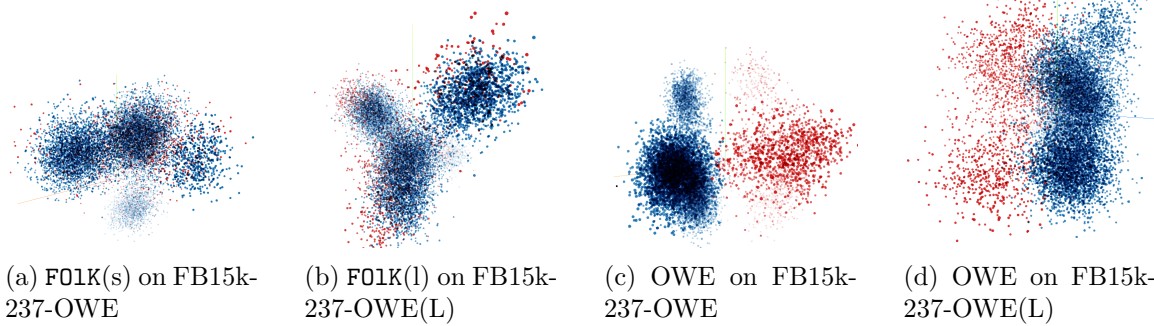

(a) FO1K(s) on FB15k-237-OWE

(b) FO1K(l) on FB15k-237-OWE(L)

(c) OWE on FB15k-237-OWE

(d) OWE on FB15k-237-OWE(L)

Figure 2: Visualization of embeddings using PCA. The closed-world and projected embeddings are differentiated by red and blue dots respectively. FO1K embeddings are indistinguishable from each other. In OWE, the two embeddings cluster separately.

**RQ3: Contributions of Our Model Components** Ablation studies on FB15k-237-OWE(L) are presented in Table 6. FO1k(s) (M1,M2,M3) in the Table 6 only denotes a model choice, the dataset used is FB15k-237-OWE(L) throughout. We observe that using RoBERTa contributes to a 16% MRR improvement (M6 vs M3). Omitting $\alpha\mathcal{L}$ results in an average decrease of 35% in MRR. Similarly, omitting $\beta\mathcal{L}_{\mathcal{X}}$ results in an average decrease of 6%. The performance variation across the iterations of FO1K(l) is indicated in Tables 2, 4 and 5. The fact that performance increases uniformly (10% in Hits@10 in the case of WN18RR, Table 4) across iterations lends strong support to iterative phasewise alignment.

## 7. Acknowledgements

Balaraman Ravindran was partly supported by a research grant from Intel Research. We'd like to thank Ujjawal Soni for his contributions and Priyesh Vijayan for his valuable suggestions. We are very grateful to everyone who reviewed drafts of the paper.

## 8. Conclusion

We present FO1K - a framework to generate embeddings for open-world KG entities from their descriptions. We demonstrate that our framework significantly outperforms SOTA baselines on a number of tasks by upto 72%. In the future we would like to explore a more principled metric for measuring neighbourhood alignment across nodes and use this to improve embedding induction performance.

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

# 9. Appendix

## 9.1 The Energy and Loss functions we use in our model

Table 8: The Energy and Loss functions we use in our model

| Embedding Method | Energy Function ($\mathscr{E}$) | Loss Function ($\mathscr{L}$) |
|---|---|---|
| TransE | $\mathscr{E}(h,r,t) = \|\boldsymbol{h} + \boldsymbol{r} - \boldsymbol{t}\|_2$ | $\mathscr{L} = \sum\limits_{(h,r,t)\in\mathcal{T}} \sum\limits_{(h^\circ,r^\circ,t^\circ)\in\mathcal{T}^{corr}} [\alpha + \mathscr{E}(h,r,t) - \mathscr{E}(h^\circ,r^\circ,t^\circ)]_+$ |
| DistMult | $\mathscr{E}(h,r,t) = \langle\boldsymbol{h},\boldsymbol{r},\boldsymbol{t}\rangle$ | $\mathscr{L} = \sum\limits_{(h,r,t)\in\mathcal{T}^\circ} log(1 + exp(-I_{h,r,t}\mathscr{E}(h,r,t)))$ |
| ComplEx | $\mathscr{E}(h,r,t) = Re(\langle\boldsymbol{h},\boldsymbol{r},\overline{\boldsymbol{t}}\rangle)$ | $\mathscr{L} = \sum\limits_{(h,r,t)\in\mathcal{T}^\circ} log(1 + exp(-I_{h,r,t}\mathscr{E}(h,r,t)))$ |

## 9.2 Comparing properties of open-world KG completion methods

Table 9: The properties of different open-world KG completion methods

|  | DKRL | JointE | ConMask | OWE | FO1K |
|---|---|---|---|---|---|
| Embedding |  | ✓ |  | ✓ | ✓ |
| Joint | ✓ | ✓ | ✓ |  | ✓ |
| Efficient ranking | ✓ | ✓ |  | ✓ | ✓ |
| Modular |  |  |  | ✓ | ✓ |
| Sequence size aware |  |  |  |  | ✓ |

## 9.3 Loss Terms — Intuition

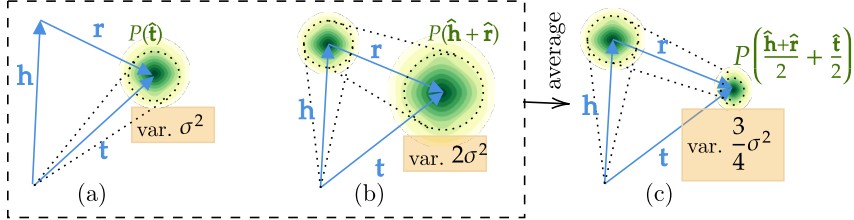

Figure 3: The probability contour plots for $\hat{\boldsymbol{h}}$, $\hat{\boldsymbol{r}}$ and $\hat{\boldsymbol{t}}$ are shown in green. Fig. (a) depicts the variance around $\boldsymbol{t}$ using only $\hat{\boldsymbol{t}}$ as an estimate. Fig. (b) depicts the variance around $\boldsymbol{t}$ using only $\hat{\boldsymbol{h}} + \hat{\boldsymbol{r}}$ as an estimate. Fig. (c) depicts the variance around $\mathbf{t}$ using an average of the two as the true estimate.

We explain why we need additional loss terms when they seem to enforce the same objective as $\mathscr{L}_{proj}$. We argue that in a stochastic setting, the three loss terms' summation gives us the lowest variance estimate of the ideal embeddings. The reasoning is along the same lines as how bagging reduces model variance as long as the errors are uncorrelated. We explain the

intuition behind this through Fig. 3 and demonstrate the effectiveness of the cross-domain tasks empirically in Table 6.

Let's consider for simplicity, the translational framework of TransE. Consider three embeddings - $\hat{\boldsymbol{h}}, \hat{\boldsymbol{r}}, \hat{\boldsymbol{t}}$ normally distributed around the *ideal* embeddings - $\boldsymbol{h}, \boldsymbol{r}, \boldsymbol{t}$ with approximately equal diagonal variance $\sigma^2$. During training we'd like to align the projected embedding tail to the ideal tail embedding **t**. If we minimise $\mathscr{L}_{proj}$ on its own we use $\hat{\boldsymbol{t}}$ as an estimate of $\boldsymbol{t}$, resulting in a variance of $\sigma^2$ around the true tail. Similarly using $\mathscr{L}_X$ gives us a variance of $2\sigma^2$. Averaging the two estimates, i.e. minimising $\mathscr{L}_X + \mathscr{L}_{proj}$ gives us the lowest variance - $3/4\sigma^2$ (follows from the addition of uncorrelated gaussian random variables), *provided* $\hat{\boldsymbol{h}} + \hat{\boldsymbol{r}}$ and $\hat{\boldsymbol{t}}$ are uncorrelated. We observe that this is the case empirically. The mean absolute correlation between any two dimensions of the two vectors starts off at 0.02 and tapers off at 0.07

### 9.4 Structural Embeddings — Hyperparameters

We use pre-trained structural embeddings as an initialization for FO1K(l). For this, we only perform a hyperparameter search on ComplEx. Our hyperparameter choice here is based on the best `MRR` performance on the closed-world validation set and so, can be computed efficiently using fast KG embedding libraries like OpenKE[5]. For many popular KG datasets and embedding models, these values are publically available. We don't finetune these parameters to improve open-world performance. For training ComplEx embeddings, as recommended by the original paper we vary the learning rate in {1.0, 0.5, 0.1, 0.05, 0.01}, the number of negative samples in {1,10,100} and $\lambda$ in {0, 0.1, 1}.

We use 300-dimensional embeddings throughout. We use the same optimal parameters for FO1K(s). During the training of FO1K(s) and subsequent iterations of phase 1 of FO1K(l), we reduce the number of negative samples for WN18RR-Open and YAGO3-10-Open to 10 for computational efficiency.

Table 10: Hyperparameter settings for training the structural embeddings of ComplEx on different datasets.

| Dataset | Learning Rate | # -ve samples | $\lambda$ |
|---|---|---|---|
| WN18RR-Open | 0.01 | 100 | 1 |
| FB15k-237-OWE | 0.03 | 1 | 0 |
| YAGO3-10-Open | 0.05 | 100 | 0.1 |

We take the optimal hyperparamters for FB15k-237-OWE on ComplEx, DistMult and TransE from the implementation of OWE. TransE's margin parameter is set to 1.

Table 11: Hyperparameter setting for different structural models on FB15k-237-OWE

| Structural Model | Learning Rate | # -ve samples | $\lambda$ |
|---|---|---|---|
| TransE | 0.1 | 1 | 0 |
| DistMult | 0.01 | 1 | 0 |
| ComplEx | 0.03 | 1 | 0 |

---

5. https://github.com/thunlp/OpenKE

### 9.5 Closed-World Performance:

The closed-world KG completion performance of FO1K compared to the performance of the base KG embedding method, ComplEx is presented in Tables 3 and 12. Rather than affecting closed-world performance, joint training elicits an average MRR improvement of 5%. We observe that this improvement is largely due to entities missing key relations in the KG. Training jointly with descriptions acts as a regularizer, ensuring these entities are embedded in correct local neighbourhoods, despite the missing relations.

Table 12: Performance on closed-world KG completion compared with our base KG embedding model, ComplEx. All of the datasets in this comparison have long descriptions.

| | YAGO3-10-Open | | | | WN18RR-Open | | | | FB15k-237-OWE(L) | | | |
|---|---|---|---|---|---|---|---|---|---|---|---|---|
| Model | MRR | H@1 | H@3 | H@10 | MRR | H@1 | H@3 | H@10 | MRR | H@1 | H@3 | H@10 |
| ComplEx | 40.4 | 31.8 | 45.0 | 57.2 | 48.5 | **45.8** | 50.0 | 53.5 | 35.6 | 24.3 | 41.2 | 57.3 |
| FO1K(l) iter. 2 | **41.2** | **32.3** | **46.2** | **60.8** | **50.4** | 45.5 | **50.4** | **55.6** | **37.6** | **27.5** | **43.6** | **59.3** |

### 9.6 Geometric Properties

To analyse the geometric properties for alignment across structure embedding space and projected embedding space, we conduct experiments along the lines of Ormazabal et al. [Ormazabal et al., 2019]. We calculate two metrics – eigen value similarity and hubness. We centre and normalise both the structural embeddings and the projected embeddings. We contruct graph laplacians for the two spaces and identify $k_1$ and $k_2$, the minimum number of eigen values that account for 90% of the sum of all eigen values in either space. We then calculate the average squared difference between the top $min(k1, k2)$ eigen values between the two spaces. Hubness is the minimum % of induced embeddings that are the nearest neighbours to at least N% of structural embeddings. We report hubness at 10%. A larger value indicates lower hubness which is preferable.

Table 13: Comparing the geometric properties of joint and offline methods. We use ↑ to indicate that a higher value is better and vice-versa. We use FO1K(s) for FB15k-237-OWE and FO1K(l) for FB15k-237-OWE(L)

| | FB15k-237-OWE | | FB15k-237-OWE(L) | |
|---|---|---|---|---|
| | Eig. Sim.↓ | Hub. 10%↑ | Eig. Sim.↓ | Hub. 10%↑ |
| OWE (Offline) | 998 | 0.2 | 34981 | 0.3 |
| FO1K (Joint) | **15.3** | **1.3** | **1.5** | **0.8** |

## 9.7 Ranking Performance

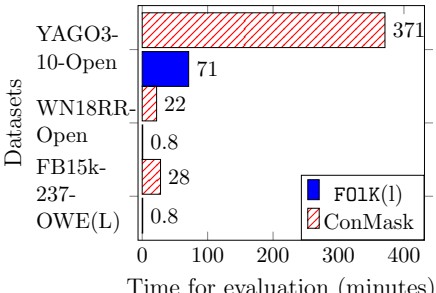

Figure 4: Time taken to rank triplets by ConMask and FO1K(l) across various datasets.

