# OpenReview forum: "A Joint Training Framework for Open-World Knowledge Graph Embeddings"
_AKBC.ws/2021/Conference — AKBC 2021_

### Official Review · Reviewer_Rvrn · 2021-07-20
**The paper is a decent focused contribution. However, the experimentation and the writing can be significantly improved.**

**Rating:** 5
**Confidence:** 3

**Review:**

This paper presents an approach to learn representations of (open-world) entities in a KG with given textual descriptions of open-world entities. This paper describes a technique that jointly learns embeddings for KG entities from descriptions and KG structure for open-world knowledge graph completion. The technique is experimentally validated on the YAGO3-10-Open and WN18RR-Open datasets and it beats previous open world entity representation learning methods.

I think this paper is a decent focused contribution. However, I think it can be significantly improved in its presentation quality:
1. The drawbacks of existing approaches is not clearly described and thus it is hard to understand the contributions of this work
2. The method description is not complete. For example L_proj is not described. It is hard to understand the approach.
3. The descriptions of the DSA score is also fuzzy. It is not clear how I would implement this with just this description.

The method is taken from BiSkip. But it is hard to understand how the application is different here.

Statistical significance numbers should help in table 4.

I also have the following questions:
1. Why do we need a 2 phase learning approach. How does it compare to a one phase learning model?
2. how do you find the optimal value of alpha?

---

> ### Author Response · Authors · 2021-07-30
> **Addressed all the comments and have detailed few sections in the paper as per recommendation**
>
> We are happy that the reviewer finds our contribution focused. We address R3’s concerns below while pushing back on the idea that our work needs major changes.
>
> “However, the experimentation and the writing can be significantly improved. ”
>
> We would like to point out that both reviewers R1 and R2 acknowledge that our experiments are comprehensive, the improvements over the baselines are clear and that the differences from earlier methods are clearly explained. Below we address the few concrete comments by R3.
>
> We acknowledge that the method section is a point of concern shared by R2 and have made use of the additional space to clarify key details. We also address R3's concerns about differences with previous approaches while pointing out that both R1 and R2 find the differences clearly explained.
>
> “The drawbacks of existing approaches is not clearly described and thus it is hard to understand the contributions of this work”
>
> We discuss the limitations of existing approaches in section 2. We go into further limitations in Sec. 6.3 RQ1. To make things even clearer, we have now added a table 9 to the supplementary materials that clarifies the differences further.
>
> “For example $L_{proj}$ is not described”
>
> We have clarified $L_{proj}$ in the revision to the paper. $L_{proj}$ is the $\ell_{2}$ distance between the projected structural embeddings of the entities with their corresponding description embeddings
>
> “The descriptions of the DSA score is also fuzzy”
>
> We have clarified the DSA score in our revision
>
> “The method is taken from BiSkip. But it is hard to understand how the application is different here.”
>
> We think it is a mischaracterization to say that we have 'taken'  our method from BiSkip. BiSkip is used for a completely different task- that of bilingual lexicon induction and the algorithms are completely different. We only bring up BiSkip to indicate that the practice of using cross-modality where we minimise more than just the paired projection loss has support in other domains. We have now expanded upon the section to further clarify things.
>
> “Statistical significance numbers should help in table 4.”
>
> We outperform all our baselines on all metrics. The differences between FOlK and our baselines range from 16.6% at the low end to 86.1% at the high end. The test sets in question have between 3000 and 39000 examples. Given the magnitude of the difference and the size of the test sets, we didn't feel the need to report statistical significance numbers
>
> “Why do we need a 2 phase learning approach. How does it compare to a one phase learning model?”
>
> We report exactly this in tables 2, 4 and 12.  Wherever we report the results of FOlK(l), we also include the performance improvement across iterations mentioned in Section 6.1. The motivation behind using a 2-phase model is to allow the KG structure and entity descriptions to inform each other. This is elaborated in the 'Joint Training' section of our introduction. We also explain the need for a phase-wise training approach in section 4.4. We find that empirically this is beneficial for both open-world (Tables 2 and 4) and closed-world (Table 12) performance.
>
> “how do you find the optimal value of alpha?”
>
> We performed a geometric search and varied alpha in [1, 0.1, 0.01, 0.001]. We find that a value of 0.001 works well across all datasets. We have now added this detail to the paper.
>
> We request the reviewer to revise their rating if they see fit. We are happy to address additional concerns.

---

### Official Review · Reviewer_mGET · 2021-07-21
**Interesting fusion of text and conventional KG Embeddings, consistent improvements over baselines for all datasets/metrics**

**Rating:** 6
**Confidence:** 3

**Review:**

The authors propose a new method for computing Knowledge Graph Embeddings on KGs where textual descriptions of entities are available. They evaluate the effectiveness of their method on a number of downstream tasks, including open-world KG completion, closed-world KG completion and entity classification. They show that their method (FOlK) outperforms baselines across all tasks in key metrics.

The authors also perform several ablations to show effectiveness of different parts of their model. They also highlight areas in which their model is better than baselines such as OWE, and provide intuition for it.

However, there are several shortcomings in the way the paper is written and organized. Some of these are:

1. 'Cross domain alignment' Sec 4.3 is hard to understand. Authors say they 'treat an entity’s immediate graph neighbourhood as its context and use skipgram objective to maximize the probability of the entity given the neighbourhood entities'. However, this is nowhere reflected in the loss functions used. Loss function $\mathscr{L}$ as well which is used in eqn 2 and 3 hasn't been defined in the main text which makes things even more confusing. $\mathscr{L_{proj}}$ is not described anywhere. Training algorithm should be part of main text, since it is a vital differentiator from OWE.

2. DSA score is hard to understand, given description is not enough to be able to reproduce it.

3. 'FOlK is the first framework to use a transformer-based encoder to embed open-world entities'
KG-BERT (2019) uses BERT for encoding closed-world entities. Extending it to open-world entities is a straight forward solution and not a novelty.

4. 'the score for any triple must be of the order of the embedding dimensions'  Section 1, point 3-'Efficient Ranking'. What does this mean?

5. Whats the difference between FOlK(s) and FOlK(l)? It's not specified

6. A lot of the content that is being referred to in the main text lies in the supplementary material. The authors should move at least the most relevant stuff (such as training algorithm, loss functions) to main text.

It seems that the paper has been written in a hurry and is lacking sufficient description for the method, given that the main contribution is the proposed new method FOlK. A better method section and reorganizing is needed.

---

> ### Author Response · Authors · 2021-07-30
> **Made significant changes to clarify method section with additional space**
>
> We thank the reviewer for their comments. We are encouraged that R2 finds our improvements over the state-of-the-art baselines consistent. We are also happy that R2 has taken note of our efforts to delineate our differences from the baselines clearly.
>
> We address R2’s concerns below:
>
> 1. We acknowledge the comment and have edited Sec. 4.3 to improve clarity. In BiSkip, the authors
> use a skipgram objective where the probability of a target language word given the source language context is maximised and vice versa. We cite BiSkip to support our decision to include cross-modality objectives in addition to paired $\ell_{2}$ projection loss. Since we use the same definition for $\mathscr{L}$ as other closed-world models, we thought defining it explicitly in the supplementary materials was sufficient. Given the additional space, we have now moved the definition of the loss term used by complex to the main text. With the additional space, we have now added a definition of $\mathscr{L_{proj}}$ and training algorithm to the revised version of the paper (Section 4.3).
>
> 2. We have added more details about the DSA score (Section 5) to the revised version
>
> 3. We strongly disagree that extending transformers to open-world entities is straightforward for two reasons. (1)Unlike KG-BERT, in the open-world embedding generation problem we are working with two modes of information - text and KG structure. Aligning the two is a significant challenge that we study in detail in this work. (2)KG-BERT isn’t a KG embedding technique. We outline the differences between KG Completion and embedding a KG entity in the introduction. KG-BERT only performs the task of KG completion. So, methods like KG-BERT don't produce entity embeddings that can be used by downstream tasks. Also, using large language models at inference instead of using embeddings is very inefficient (Section 6.3 RQ1).
>
> 4. By 'order of the embedding dimensions' we are referring to the time complexity of scoring a triple. That is, that the time complexity must be O(d) where d is the dimension of the entity and relation embeddings. Models like FOlK that use a simple embedding scoring function at inference time can perform entity ranking that is several times faster (22x as noted in Sec. 6.3 RQ2) than models like ConMask that require large language models to score the likelihood of a triple. The ranking performed by models like ConMask isn't O(d) but rather depends on the language model and the size of the descriptions. This is a very important difference as ranking is a fundamental operation in KGs, that we have to perform every time we want to discover a new connection / add a new entity to our KG. By 'efficient ranking' we simply mean ranking that is faster and has lower peak-memory requirements. We have modified the Introduction section to make this clearer.
>
> 5. In section 4.2 we indicate that we use s & l to indicate the short and long descriptions. FOlK(l) is the transformer-based model we use for long descriptions and FOlK(s) is the CBOW-based model we use for short descriptions. We have now added information to Section 4.3 that makes the notation explicit.
>
> 6. We acknowledge that a lot of content lies in the supplementary material. However, we disagree that the paper was written in a hurry. This work has gone through several careful revisions. We made a conscious decision to abridge certain definitions and move less important information to the supplementary section to meet AKBC’s 10-page limit.
>
> We thank R2 for their suggestions. With the additional page that has now been allowed, we have been able to make the method section much clearer and have added the algorithm in main paper. We respectfully request that the reviewer give the new method section a read and revise their score if they see fit.

---

### Official Review · Reviewer_NgVF · 2021-07-22
**Interesting approach for jointly learning structural and textual embeddings for KGs, comprehensive experiments and analysis**

**Rating:** 7
**Confidence:** 3

**Review:**

The paper proposed a joint approach to learn embeddings for KG entities from both descriptions and KG structure, for the task of open-world knowledge graph completion. The main idea is separate modules for obtaining structural embeddings (of entities and relations in a KG) and description embeddings (of descriptions of possibly new entities), and a description projection module to project the description to a space where the structure and description embeddings are aligned (KG embeddings).

(+) The approach was evaluated on three different datasets, showcasing the robustness of the approach.

(+) Comprehensive analysis of the proposed approach compared with several prior work on the topic.

(+) The differences with considered baselines are well explained.

(-) Some design choices of the approach need justification (see Q1).

Questions:
1. What is the disadvantage of using transformer-based model to represent an entity description when the description is short (ten of fewer words)? Why is there a need to use two different encoders depending on the description length?
2. In Section 6.2, four tasks were introduced but the last one was without any elaboration unlike the other three?

---

> ### Author Response · Authors · 2021-07-30
> **A note of thanks, addressed the concerns, have clarified and included experiment on geometric properties**
>
> We thank the reviewer for their insightful and positive comments. We are glad that R1 finds our experiments and analysis comprehensive. Collating results from multiple disparate KG codebases was a considerable challenge and we are encouraged that R1 finds our comparisons adequate. We are also pleased that R1 finds our differences with existing state-of-the-art baselines well explained.
>
> We address R1’s questions below:
> 1. We already report the results of using a transformer for short descriptions in Table 2. The simpler single-phase approach results in empirical improvements on most metrics. In addition to the modest empirical improvement, the simpler approach converges faster (8hrs vs 48hrs) and has a smaller peak memory requirement. For these reasons, we think it is important to modify our algorithm to adapt to different KG datasets.
>
> 2. We partly studied geometric properties in Figure 2 of the original paper. To avoid confusion, we have now removed the reference to this fourth task. We have also added an additional experiment to the supplementary materials that study the geometric properties of the embeddings explicitly.

---

### Author Response · Authors · 2021-07-30
**We have addressed all concern of reviewers and have included most of the suggestions in revised paper**

We thank all the reviewers for their useful suggestions. We are happy that the reviewers find our experiments and analysis complete, our improvements clear and our method robust across datasets. We are also encouraged that most reviewers find our differences from our baselines clearly explained.

We are glad that most of the concerns brought up have to do with the clarity of the method section. In our original submission, we ran up against the 10-page limit. Given the additional page, we’ve managed to incorporate most if not all of the suggestions brought up by the reviewers.

---

### Decision · Program_Chairs · 2021-08-18

**Decision:**

Accept

**Comment:**

This paper presents a new approach called FOIK to learn entity embeddings that exploits not just the KG structure but also the textual entity descriptions, evaluated on the task of open-world knowledge graph completion. Extensive experiments show the value of the approach.
The authors were able to address most reviewer feedback in the revised version that was uploaded to the system.